# Aging Triggers H3K27 Trimethylation Hoarding in the Chromatin of *Nothobranchius furzeri* Skeletal Muscle

**DOI:** 10.3390/cells8101169

**Published:** 2019-09-28

**Authors:** Chiara Cencioni, Johanna Heid, Anna Krepelova, Seyed Mohammad Mahdi Rasa, Carsten Kuenne, Stefan Guenther, Mario Baumgart, Alessandro Cellerino, Francesco Neri, Francesco Spallotta, Carlo Gaetano

**Affiliations:** 1National Research Council, Institute for Systems Analysis and Computer Science, 00185 Rome, Italy; chcencioni@gmail.com; 2Department of Genetics, Albert Einstein College of Medicine, Bronx, NY 10461, USA; 3Leibniz Institute on Aging - Fritz Lipmann Institute (FLI), 07745 Jena, Germany; Anna.Krepelova@leibniz-fli.de (A.K.); Mahdi.Rasa@leibniz-fli.de (S.M.M.R.); Mario.Baumgart@leibniz-fli.de (M.B.); francesco.neri@leibniz-fli.de (F.N.); 4ECCPS Bioinformatics and deep sequencing platform, Max Planck Institute for Heart and Lung Research, 61231 Bad Nauheim, Germany; carsten.kuenne@mpi-bn.mpg.de (C.K.); Stefan.Guenther@mpi-bn.mpg.de (S.G.); 5Laboratory of Biology (Bio@SNS), Scuola Normale Superiore, c/o Istituto di Biofisica del CNR, 56124 Pisa, Italy; alessandro.cellerino@gmail.com; 6Department of Oncology, University of Turin, 10060 Candiolo (TO), Italy; francesco.spallotta@unito.it; 7Candiolo Cancer Institute, FPO-IRCCS, 10060 Candiolo (TO), Italy; francesco.spallotta@unito.it; 8Laboratory of Epigenetics, Istituti Clinici Scientifici Maugeri IRCCS, 27100 Pavia, Italy; carlo.gaetano@icsmaugeri.it

**Keywords:** aging, skeletal muscle, sarcopenia, frailty, chromatin, epigenetic changes, histone modifications, *Nothobranchius furzeri*

## Abstract

Aging associates with progressive loss of skeletal muscle function, sometimes leading to sarcopenia, a process characterized by impaired mobility and weakening of muscle strength. Since aging associates with profound epigenetic changes, epigenetic landscape alteration analysis in the skeletal muscle promises to highlight molecular mechanisms of age-associated alteration in skeletal muscle. This study was conducted exploiting the short-lived turquoise killifish *Nothobranchius furzeri* (*Nfu*), a relatively new model for aging studies. The epigenetic analysis suggested a less accessible and more condensed chromatin in old *Nfu* skeletal muscle. Specifically, an accumulation of heterochromatin regions was observed as a consequence of increased levels of H3K27me3, HP1α, polycomb complex subunits, and senescence-associated heterochromatic foci (SAHFs). Consistently, euchromatin histone marks, including H3K9ac, were significantly reduced. In this context, integrated bioinformatics analysis of RNASeq and ChIPSeq, related to skeletal muscle of *Nfu* at different ages, revealed a down-modulation of genes involved in cell cycle, differentiation, and DNA repair and an up-regulation of inflammation and senescence genes. Undoubtedly, more studies are needed to disclose the detailed mechanisms; however, our approach enlightened unprecedented features of *Nfu* skeletal muscle aging, potentially associated with swimming impairment and reduced mobility typical of old *Nfu*.

## 1. Introduction

In the last two centuries, life expectancy in Western countries has increased considerably, dramatically enlarging the elderly people population. Consequently, the cost of healthcare has raised significantly in association with the multi-morbidity and chronicity that characterize elderly patients [1,2,3]. By 2050, more than one fifth of the global population will be older than 60 years, and about 5% will be over 80 [1,2,3]. Accordingly, the risk of developing aging-associated diseases, including cardiovascular disease, dementia, and cancer, will continue to rise [4]. This situation challenges health and social systems, with a significant economic impact worldwide. 

Aging is accompanied by an overall loss of fitness and a decline of cell and organ function, associated with a reduced regenerative capacity and life expectancy [5]. In this context, the understanding of aging-associated molecular mechanisms is an unmet need to test novel approaches able to promote healthy aging and improve quality of life in the growing elderly population. 

Among the common features of aging, elderly people often experience loss of muscle mass, leading in some cases to sarcopenia, which associates with frailty and impaired health span [6]. Sarcopenia is present in about 10% of people older than 60 years with weakened muscle strength and impaired physical activity and mobility [6,7]. Of note, chronic inflammation and mitochondrial dysfunction have been shown to contribute to the development of sarcopenia [8,9]. Although loss of muscle mass and strength is a common feature of aging [10], the underpinning molecular mechanisms have been poorly characterized and, at present, the only effective prevention is represented by an extensive lifestyle change based on balancing physical workout and diet [9]. 

Physiologically, the aging process is characterized by profound epigenetic changes, which broadly affect the genome architecture and the epigenetic landscape, ultimately altering gene expression [2,3]. At least in part, epigenome control relies on DNA methylation, histone-modifying complexes, and non-coding RNA activity [11]. In particular, DNA methylation plays an important mechanistic role in aging [12], with specific CpG site methylation changes, either hyper- or hypo-methylated, being predictive of biological age in a variety of species including rodents, primates, and humans [13]. DNA methylated regions are usually enriched with specific methylated histone residues, including histone 3 (H3), lysine (K) 27, and lysine 9 tri-methylation (H3K27me3; H3K9me3) [14], which contribute to heterochromatin formation within the genome. Indeed, H3K27me3 and H3K9me3 enrichment contributes to the formation of senescence-associated heterochromatin foci (SAHFs) [15]. These are nuclear DNA domains densely stained by 4’,6-diamidino-2-phenylindole (DAPI), which associates with distinct heterochromatin structures in stress-induced senescent cells [16,17]. On the contrary, acetylation of histones is rather associated with an open chromatin configuration, transcript elongation, and therefore an active gene expression [18]. Open chromatin is characterized by regions enriched in histone 4 (H4), K16 and H3K9 acetylation (ac) (H4K16ac; H3K9ac) [19,20]. 

During aging, the histone modification landscape change considerably influences gene expression, genomic stability, DNA repair, and replication [2,5]. In humans, global DNA and histone methylation decreases with age; however, it accumulates at specific loci [21], leading to loss of heterochromatin and genome instability [22]. Alteration in histone modifications also depends on the functional deregulation of histone methylases and demethylases, including, respectively, the Polycomb repressive complex (PRC) members EZH2 (PRC2) and Bmi1 (PRC1) or the jumonji domain containing 3 (JMJD3) and the lysine demethylase family 6 (KDM6) [23]. 

In the present study, we investigated the epigenetic landscape of young (5 weeks old), adult (12–21 weeks old), and old (27–40 weeks old) *Nothobranchius furzeri* (*Nfu*) skeletal muscle tissue to gain insights about epigenetic changes associated with aging and the related impact on chromatin structure and cellular transcriptome occurring in this model. In recent years, in fact, the turquoise killifish *Nfu* has been established as a model for aging research [24,25,26] due to its exceptionally short lifespan (3–9 months) [27] and its capacity to resemble many hallmarks of mammalian aging [28,29], such as sarcopenia [30]. Indeed, old *Nfu* present reduced locomotor activity as a consequence of age-associated changes in skeletal muscle tissue [31]. Moreover, the understanding of the aging-dependent epigenetic alterations might open the avenue for novel treatments aimed at counteracting muscle fiber loss and the onset of sarcopenia, promoting healthy ageing.

## 2. Materials and Methods

### 2.1. In Vivo Experiments

Whole skeletal muscle tissue was collected from male *Nfu* (strain MZM-04/10). *Nfu* samples were derived from fish at different ages: Young (5 weeks), adult (12–21 weeks), and old (27–40 weeks). To avoid the effects of circadian rhythms and feeding, animals were always sacrificed at 10 a.m. in fasted state. Tissue collection was performed in fishes euthanized with MS-222 and cooled on crushed ice. Then, RNA was extracted from tissues in a solution of cooled QIAzol (Qiagen), together with a 5 mm stainless steel bead (Qiagen). Homogenization was performed by TissueLyzer II (Qiagen). Proteins and DNA were extracted from flash-frozen fish samples. Immunofluorescence microscopy was performed in paraformaldehyde (PFA)-fixed tissue embedded in paraffin.

### 2.2. Cells and Treatments

Cells were isolated from *Nfu* skeletal muscle tissue minced on ice in PBS. Tissues from up to three individuals were pooled. Collagenase solution (1 mg/mL) was added to samples and incubated for 30 min on a rotation platform at 37 °C. Then, digestion step was stopped, cell suspension was filtered, and pellet was obtained by centrifugation at 350 g for 10 min at room temperature. To reduce contamination from fibroblasts, two pre-plating steps were performed. Thereafter, cells were plated in collagen-coated 6-well plates and incubated at 28 °C. Cells were cultured in DMEM medium supplemented with 10% water, 20% FBS, 1% Penicillin/Streptomycin, 0,1% Amphotericin, and 0.05% Gentamycin. Medium was changed every other day and cells were split before reaching 80–90% of confluence.

### 2.3. Quantification of Global DNA Methylation

DNA was extracted using the E.Z.N.A. DNA tissue kit (VWR OMEGA biotek) according manufacture’s instruction. MethylFlash Methylated (5mC) DNA Quantification Kit was used to quantify methylation levels of DNA from fish and cellular samples according manufacturer’s instruction (Epigentek). The optical density (OD) was detected by EnSpire Multimode Plate Reader (Perkin Elmer).

### 2.4. RNA Extraction and qRT-PCR

RNA was extracted from *Nfu* tissue and cell samples using Tri-Reagent (Sigma-Aldrich) according to manufacturer’s instruction. cDNA synthesis for quantitative real-time PCR (qRT-PCR) was carried out with SuperScript III First-Strand Synthesis Super Mix for qRT-PCR (Invitrogen) according to the manufacturer’s protocol. All reactions were performed in 96-well format in the StepOne Plus Real-Time PCR System (Applied Biosystems) using ORA qPCR Green ROX H Mix (HighQu). For each gene of interest, qRT-PCR was performed as follows: Each RNA sample was tested in duplicate and insulin receptor a (ira) was used to normalize transcript abundance in *Nfu*. mRNA expression levels were calculated by Comparative Ct Method (Pfaffl, 2004) by using the Applied Biosystems software (Applied Biosystems) and were presented as fold induction of transcripts for target genes. Fold change above 1 denotes up-regulated expression, and fold change below 1 denotes down-regulated expression versus reference sample.

Primers sequences (Table 1) were selected using the primer3 software (Untergasser et al., 2012) based on published sequence data from NFINgb (http://nfingb.leibniz-fli.de/main.html?data=data/json/notho4) [32] and the Killifish Genome Browser (http://africanturquoisekillifishbrowser.org/) [33]. All primers were synthesized by Sigma-Aldrich.

### 2.5. Sequencing and Bioinformatics Analysis

For RNA sequencing, RNA was isolated from skeletal muscle tissue from 3 fishes for each condition (young, adult, and old) using the miRNeasy micro Kit (Qiagen) combined with on-column DNase digestion (DNase-Free DNase Set, Qiagen) to avoid contamination by genomic DNA. RNA and libraries integrity were verified with LabChip Gx Touch 24 (Perkin Elmer). Next, 1µg of total RNA was used as input for SMARTer Stranded Total RNA Sample Prep Kit - HI Mammalian (Clontech). Sequencing was performed on the NextSeq500 instrument (Illumina) using v2 chemistry, resulting in average of 30 million reads per library with 2 × 75 bp paired-end setup. The resulting raw reads were assessed for quality, adapter content, and duplication rates with FastQC (available online at http://www.bioinformatics.babraham.ac.uk/projects/fastqc). Trimmomatic version 0.33 was employed to trim reads after a quality drop below a mean of Q20 in a window of 5 nucleotides. Only reads above 30 nucleotides were cleared for further analyses. Trimmed and filtered reads were aligned versus the *Nothobranchius furzeri* genome version NotFur1 using STAR 2.4.2a with the parameter “--outFilterMismatchNoverLmax 0.1” to increase the maximum ratio of mismatches to mapped length to 10%. The number of reads aligning to genes was counted with featureCounts 1.4.5-p1 tool from the Subread package. Only reads mapping at least partially inside exons were admitted and aggregated per gene. Reads overlapping multiple genes or aligning to multiple regions were excluded. Differentially expressed genes were identified using DESeq2 version 1.62.25. Only genes with a minimum fold change of ±2, a maximum Benjamini–Hochberg corrected *p*-value of 0.05, and a minimum combined mean of 5 reads were deemed to be significantly differentially expressed. The Ensembl annotation was enriched with UniProt data (release 06.06.2014) based on Ensembl gene identifiers (Activities at the Universal Protein Resource (UniProt)). The correlation of replicate gene counts was assessed with the Spearman ranked correlation algorithm included in R 3.11 (R: A language and environment for statistical computing). Genes regulated by age (±1 log2 fold change, base-mean > 5, FDR < 0.05) derived from mRNASeq of killifish Nfu samples were imported into DAVID (https://david.ncifcrf.gov/) to reveal top KEGG pathways affected by age.

For ChIP sequencing, around 20 mg of skeletal muscle tissue were chopped with a scalpel, harvested in 5 mL of PBS, cross-linked by 1.5% formaldehyde for 30 min at room temperature on a rotator, and quenched with 0.125 M glycine for 5 min at room temperature. After crosslinking, tissue was washed two times in cold PBS and centrifuged at 1000× *g* for 5 min at 4 °C. The pellet was resuspended in 0.25 mL of SDS lysis buffer (50 mM Tris pH 8.0, 1% SDS, 10 mM EDTA, anti-proteases), incubated on a rotator for 30 min at 4 °C, sonicated for 18 cycles on high power setting (30 s ON, 30 s OFF) using the Bioruptor Next Gen (Diagenode) and centrifuged at 20,000× *g* for 10 min at 4 °C. The isolated chromatin was diluted 10-fold with ChIP dilution buffer (16.7 mM Tris-HCl pH 8.0, 0.01% SDS, 1.1% Triton X-100, 1.2 mM EDTA, 167 mM NaCl) (1/10 was kept as input) and incubated with 4 µg of antibody overnight at 4 °C on a rotator. Protein G-conjugated magnetic beads (Dynal, Thermo Fisher Scientific) were saturated with PBS/1% BSA overnight at 4 °C. The next day, samples were incubated with saturated beads for 2 h at 4 °C on a rotator, and subsequently washed with 1 mL of cold low-salt buffer (20 mM Tris-HCl pH 8.0, 0.1% SDS, 1% Triton X-100, 2 mM EDTA, 150 mM NaCl), 1 mL of cold high-salt buffer (20 mM Tris-HCl pH 8.0, 0.1% SDS, 1% Triton X-100, 2 mM EDTA, 500 mM NaCl), 1 mL of cold LiCl buffer (10 mM Tris-HCl pH 8.0, 1% DOC, 250 mM LiCl, 1 mM EDTA, 1% NP-40), and twice with 1 mL of cold TE buffer (10 mM Tris-HCl pH 8.0, 1 mM EDTA). The immunoprecipitated chromatin was eluted with 200 µl of Elution buffer (10 mM Tris-HCl pH 8.0, 1 mM EDTA, 1% SDS, 150 mM NaCl, 5mM DTT) for 30 min at room temperature on a rotator, and decross-linked at 65 °C overnight. The decross-linked DNA was purified using the QiaQuick PCR Purification Kit (Qiagen) according to the manufacture’s instruction. The following antibodies were used: Rabbit anti-H3K9ac (ab4441, Abcam), rabbit anti-H3K27me3 (abe44, Millipore), and rabbit anti-IgG (Millipore). The purified ChIP DNA was end-repaired, dA-tailed, and adaptor-ligated using the NEBNext® DNA Library Prep Master Mix Set for Illumina (NEB) following the manufacturer’s instructions. The size of the library was checked using Fragment Analyzer (Agilent) and the library was sequenced on the NextSeq500 platform (illumina). Fastq files quality check was performed with FastQC (v0.11.5). Fastq files mapping to *Nfu* genome (Nfu_20150522; http://nfingb.leibniz-fli.de) was performed using Bowtie (v1.1.2) with --best --strata -m 1 parameter. Duplicate reads were removed using a custom script. For peak calling, macs14 (v1.4.2) was used with --nolambda parameter and 1e-3 as *p*-value cutoffs. The significant peaks have been used as the reference for calculation of read per million (RPM) for each sample by using a custom script.

Sequencing data sets are available on the public repository GEO: RNASeq at https://www.ncbi.nlm.nih.gov/geo/query/acc.cgi?acc=GSE135032; and ChIPSeq at https://www.ncbi.nlm.nih.gov/geo/query/acc.cgi?acc=GSE135129.

### 2.6. Western Blot

Western blotting was performed by standard procedures after tissue lysis in Laemmli buffer. Nitrocellulose blotted membranes were probed with the following antibodies: H3K27me (Abcam, ab6002), H3K9me3 (Abcam, ab8898), H4K20me3 (Abcam, ab9053), H3K4me3 (Abcam, ab8580), H3 (Cell Signalling), PCNA (GeneTex, GTX124496), γH2AX (Genetex, GTX127343), and α-Tubulin (Cell Signalling, 3873S). Development was performed by Odyssey CLX reader (LI-COR). Densitometry analysis was performed using LI-COR software. Signal intensity from three independent Western blots loaded with lysates derived from different individuals was used for densitometry calculations normalized to young samples.

### 2.7. Histology and Morphometric Analysis

Immunofluorescence microscopy and immunohistochemistry analyses were carried out according standard procedures. H3K27me3 (Abcam, ab6002), H3K9Ac (Abcam, ab4441), and HP1α (Bioss, 3825R) were used according to manufacturer’s instructions, and nuclei were counterstained with DAPI or TOPRO-3 solution. Sudan Black B staining was performed as described in [34]; no counterstaining with Fast Red was performed. Immunofluorescence was analyzed using a Leica TCS SP8 confocal microscope. Immunohistochemistry was analyzed using a Motic AE2000 light microscope (Motic Electric Group Co.).

### 2.8. Statistical Analysis

Statistical analyses were performed using GraphPad Prism 6 software. Sample sizes (*n*) were reported in the corresponding figure legend. No statistical method was used to predetermine sample size. Investigators performing sequencing analysis were blinded during the experiment. All values were presented as mean ± standard error of the mean (s.e.m.) of at least three independent experiments, unless otherwise indicated. Statistical analyses were performed using non-parametric student’s *t*-test (unpaired Kolmogorov–Smirnov test) when the comparison was done between two groups and non-parametric one-way ANOVA (unpaired Kruskal–Wallis test) for more than 2 groups. For all statistical analysis, a value of *p* ≤ 0.05 was deemed statistically significant.

### 2.9. Ethics Statement

All experiments were performed in accordance with relevant guidelines and regulations. Animals were bred and kept in FLI´s fish facility according to §11 of the German Animal Welfare Act. The protocols of animal maintenance were approved by the local authority in the State of Thuringia (Veterinaer- und Lebensmittelueberwachungsamt) with license number J-SHK-2684-04-08/11 (before August 2017) and J-003798 (since August 2017). Sacrifice and organ harvesting was performed according to §4(3) of the German Animal Welfare Act.

## 3. Results

### 3.1. Chromatin Landscape Discriminates Among Young, Adult, and Old Nfu

There is, at present, an apparent lack of information regarding epigenetic modifications in the context of age-related alteration of skeletal muscle. Moreover, very little is known about the chromatin landscape of *Nfu* skeletal muscle and how it changes during aging. Here, experiments, aimed at exploring the chromatin landscape characterizing young, adult, and old *Nfu* skeletal muscle samples, were performed on histones H3 and H4 specific tail residues. A few modifications were identified. Specifically, in old skeletal muscle tissue, we observed a significant increase in H3K27me3, H3K9me3, and H4K20me3 (Figure 1a). These modifications are associated with a closed chromatin conformation relatively abundant in heterochromatin and typically leading to gene repression. To further explore how chromatin landscape might change during aging, we extended our evaluation to histone marks more often associated with open chromatin conformation. In aged *Nfu* skeletal muscle, Western blot analysis revealed a decrease in H3K9ac and H4K16ac signals (Figure 1a). Confocal microscopy (CM) confirmed the increase in H3K27me3 (Figure 1b) and the reduction of H3K9ac (Figure 1c) in old *Nfu* compared to young skeletal muscle. Moreover, in old skeletal muscle, CM analysis of heterochromatin protein 1 alpha (HP1α), which is an additional marker of closed chromatin, provided evidence of co-localization with H3K27me3 (Figure 1d), suggesting further that old animal epigenome presents a predominantly closed conformation. These evidences prompted us to investigate the global state of DNA methylation. Interestingly, we found that with age, a steep increase of 5-methyl cytosine (5mC) occurs at adult age compared to young muscle tissue. The level of 5mC remains high at old age, maintaining statistical significance against young samples, while increasing slightly compared to adult skeletal muscle without reaching statistical significance (Figure 1e). These evidences suggest that the increase of 5mC at adult stage might contribute to the aging phenotype and to the establishment of heterochromatin. Similar results emerged in satellite cells isolated from young and old fish (Appendix A). 

Taken together, these observations show that, in aged *Nfu* skeletal muscle, chromatin becomes more condensed and possibly less transcriptionally accessible. 

### 3.2. Epigenetic Enzymes Responsible for Chromatin Condensation are Differentially Expressed Among Young, Adult, and Old Nfu

In association with the observed histone modifications, the expression of a number of relevant epigenetic enzymes responsible for heterochromatin was examined. The Polycomb repressive complex 2 (PRC2) is a protein complex with histone methyltransferase activity, which primarily leads to H3K27me3 accumulation. The mRNA levels of three important PRC2 subunits, namely ezh1, ezh2, and eed, increased in skeletal muscle tissue of old *Nfu* compared to young animals (Figure 2a). The mRNA level of PRC1 members showed a similar trend (Figure 2b). To further investigate the epigenetic enzymes involved in histone methylation, we evaluated the mRNA expression of the lysine demethylases (kdm), the enzymes involved in the removal of methyl group from lysine residues. Specifically, we focused on members of the lysine demethylase 6 (KDM6) family, which is selective for H3K27me3. Interestingly, compared to youngsters, mRNA levels of kdm6a and kdm6b were significantly reduced in adult *Nfu* muscle tissue; however, no changes of markers associated with young tissues could be detected in older animals (Figure 2c). Further studies are necessary to investigate this interesting aspect. Nevertheless, these findings suggest that increased levels of methylated histones might be the result of an increase in PRC members paralleled by partial reduction in KDM6 expression occurring during adulthood.

### 3.3. RNA Sequencing of Nfu Skeletal Muscle Tissue Shows Age-Specific Expression Pattern

To gain insight into aged-associated alterations in the *Nfu* we explored the transcriptome of young, adult, and old skeletal muscle tissue by RNA sequencing. In this experiment, analyzing *Nfu* at different ages, 5008 differentially expressed genes (DEGs) were found. Specifically, after pairwise comparison of young/adult, young/old, and adult/old, 4271, 3074, and 453 genes were found differentially expressed with more than ±1 log2 fold change (base-mean ≥ 5, FDR ≤ 0.05), respectively.

Based on multiple testing-adjusted *p*-value criteria, the 50 most significantly DEGs of each pair (young/adult, young/old, adult/old) were selected, resulting in 122 different DEGs (Appendix A). DESeq-normalized counts of the selected sequences were averaged per condition and depicted as a heatmap by using a hierarchical clustering generated by Pearson correlation of the z-score (Figure 3a). The result revealed that the aging process determined important changes in the *Nfu* skeletal muscle transcriptome. Further, the partially overlapping DEGs among the experimental contrasts were utilized to identify age-specific regulated genes, and a Venn diagram was created to assemble RNAs into up- and down-regulated genes (Figure 3b). We identified 1063 genes down-regulated and 1138 up-regulated in adults compared to young (Figure 3b,c). The old/young comparison identified 403 down-regulated and 480 up-regulated genes (Figure 3b,c). To assign a putative role to these aging-associated genes, KEGG pathway analysis was performed on significantly down- or up-regulated groups of genes (Figure 3d). The interconnections among down-regulated genes suggests that the related molecular pathways are associated with cell cycle progression and DNA repair. On the contrary, the most up-regulated genes were associated with inflammation and metabolism, particularly glycolysis (Figure 3d). This finding is typical of senescent cells and often coexists with DNA damage and the alteration of other aging-dependent regulatory pathways, including inflammation. 

Altogether, these results suggest that, similar to mammalians, in *Nfu*, the progression of aging leads to chromatin condensation associated with regions transcribing for genes involved in proliferation and DNA repair.

### 3.4. H3K27me3 and H3K9ac Show Opposite Regulation and Role During Aging

To better define the role of the observed aging-dependent histone modification alterations on DEGs, we performed a chromatin immunoprecipitation-sequencing (ChIPSeq) experiment for H3K27me3- and H3K9ac-enriched regions in the skeletal muscle of young and old *Nfu*. In aged animals, ChIP performed with the H3K27me3 or the H3K9ac antibody showed an accumulation of H3K27me3-enriched regions (Appendix A). After sequencing, we mapped the reads on the *Nfu* genome and a correlation analysis of the mapped reads revealed that H3K27me3 enriched samples clustered independently from H3K9ac samples (Appendix A). As expected, ChIP-seq indicated that H3K27me3 was particularly enriched on repetitive elements and promoters of low expressed genes with higher density around the transcriptional start site (TSS). Conversely, H3K9ac signals were more abundant on the promoters of expressed genes exhibiting a sharper distribution profile around TSS (Appendix A). Large genomic views of the mapped reads confirmed the different distribution anticipated by ChIP-seq reads. These findings are complementary, backing the experimental accuracy (Appendix A).

To understand whether these two epigenetic modifications correlate with the transcriptional alterations observed during aging (Figure 3), we calculated the ratio between H3K27me3 and H3K9ac signal for a subset of DEGs in old versus young *Nfu*. Except few cases (e.g., Dnmt3a), most of the down-regulated genes showed an aging-dependent increase of H3K27me3 and a relative decrease of the H3K9ac signal on their promoters. On the contrary, up-regulated genes showed an aging-dependent decrease of H3K27me3 and a relative increase of the H3K9ac signal (Figure 4a and Appendix A). The old/young signal ratio of H3K27me3 ChIP-seq was significantly inversely correlated (Pearson = −0.57) compared to the H3K9ac ratio, suggesting that these two histone modifications could have an opposite but synergic role during aging (Figure 4b). Genomic views of the ChIP-seq mapped reads confirmed the opposite loss and gain of these two histone modifications in specific epigenomic regions (highlighted in yellow) and particularly around TSS (Figure 4c,d). 

These results are consistent with the functional association between aging and condensed chromatin conformation in *Nfu* skeletal muscle.

### 3.5. Impairment of Cell Cycle, Differentiation, and DNA Repair Mechanics in Old Nfu Skeletal Muscle Tissue

The skeletal muscle tissue of old *Nfu* was further characterized by Western blot and qRT-PCR to assess the expression of genes involved in cell cycle, differentiation, and DNA repair. The mRNA expression of cyclin B1 (CCNB1), fundamental for the transition through mitosis, and of cyclin D1 (CCND1), driving G1/S transition, significantly decreased in old *Nfu* (Figure 5a). Consistently, the expression of proliferating cell nuclear antigen (PCNA), important for DNA replication and chromatin remodeling, was reduced in old *Nfu* muscle tissue both at the mRNA and the protein levels (Figure 5b). Intriguingly, the decrease of markers associated with proliferation was paralleled by a reduction of the muscle-specific transcription factor, myogenin-g (myog), which coordinates skeletal muscle development and repair (Figure 5c). Moreover, in agreement with RNA-seq analyses showing an impairment of genes involved in DNA repair, we observed a reduction in the mRNA level of the growth arrest and DNA-damage-inducible protein (gadd45γ) responsible of cellular stress response (Figure 5d). Worthy of note, in other model systems, low levels of gadd45γ have been associated with several types of tumors [35,36]. To further explore the status of DNA damage accumulation in old *Nfu* skeletal muscle, we analyzed the presence γH2AX, a marker of double-strand DNA breaks, which contributes to nucleosome formation, chromatin remodeling, and DNA repair [37]. As expected, we found γH2AX enriched in old *Nfu* skeletal muscle (Figure 5e). Again, similar results were found in satellite cells isolated from old *Nfu* muscle tissue (Appendix A). 

These evidences support the bioinformatics analyses of sequencing data set performed on aging *Nfu*. 

### 3.6. Up-Regulation of Inflammation and Senescence in Old Nfu Skeletal Muscle Tissue

Since the pairwise comparison between skeletal muscle tissue of old and young *Nfu* pointed out an increase in genes involved in inflammation response, we analyzed the expression of a series of genes actively participating in immune and inflammation response. Specifically, we found an increase in the mRNA levels of important genes, such as the CCAAT/enhancer-binding protein beta (Cebpb), a transcription factor regulating the expression of immune and inflammatory response related genes [38]; the janus kinase 3 (Jak3), coupled to cytokine receptors and responsible for their signaling transmission [39]; and the pentraxin-related protein 3a (Ptx3a), a molecule released from dendritic cells, fibroblasts, and endothelial cells in response to primary inflammation stimuli [40] (Figure 6a). These evidences emerged from our OMICs studies about age-specific alteration of the skeletal muscle in *Nfu* and are provided here as examples of genes affected by the increased condensation of the chromatin observed during *Nfu* aging. Interestingly, a sustained inflammatory response often associates with senescence and aging. Then, we investigated the degree of senescence in old *Nfu* skeletal muscle tissue. Although we are aware that senescence-associated-β-galactosidase activity (SA-β-gal) is the gold standard to reveal senescent cells, the use of this specific staining is limited to fresh samples, whereas it cannot be performed in formalin-fixed paraffin-embedded tissues. For this reason, we took advantage of histochemical staining with Sudan Black B (SBB) that specifically stains lipofuscin, an aggregate of oxidized proteins, lipids, and metals, known to accumulate in aged tissues, and is detectable independently of sample preparation [34,41]. Specifically, the histological staining of formalin-fixed tissue by SBB showed evident signs of senescence in old *Nfu* skeletal muscle (Figure 6b, see black arrows). Similar results were obtained analyzing the satellite cells isolated from *Nfu* skeletal muscle for β-galactosidase (β-gal) and p21 mRNA level (Appendix A). The positivity to senescent-specific staining was paralleled by an increase in the mRNA levels of p21, the cyclin dependent kinase inhibitor 1, a well-known senescence marker (Figure 6c). Actually, p21 expression increased with statistical significance in adult and old fish compared to young samples. Instead, the apparent difference between adult and old fish was not statistically significant; p21 expression in fact, does not significantly change in old fish, rather its expression remains elevated since adulthood. This evidence suggests that the increase of p21 in an adult fish might promote aging. Notably, different from mammalians, in *Nfu*, p16 cannot be used as a senescence marker. Indeed, contrasting data are present in the literature about p16 expression in teleosts. Previous manuscripts, in fact, described that no changes occur in p16 during aging [42,43]. For this reason, we focused on p21 expression. 

To gain insight into senescence, we generated a heatmap showing the most recognized senescent associated markers, including Lamin B1, HMGB1, IL-6, and IL-8 (Figure 6d). According RNA sequencing data set, we found that Lamin B1 was down-modulated about –2.29-fold in old compared to young fish. This observation is in agreement with the reported reduction of Lamin B1 typically observed during senescence and aging [44]. As regards other markers, HMGB1 was only slightly up-regulated during aging by about 0.85-fold. Although IL-6 was not detected by our sequencing, its signaling transporter (IL6ST) and receptor (IL6R) were both significantly up-regulated during aging. Finally, IL8 was up-regulated about 1.86-fold in old compared to young animals. Moreover, the increased signs of senescence and the metabolic imbalance towards glycolysis, extrapolated from RNA-seq bioinformatics, prompted us to analyze the mitochondrial function. Indeed, senescent cells accumulate dysfunctional mitochondria, characterized by decreased oxidative phosphorylation efficiency, which in turn leads to a higher production of reactive oxygen species contributing to DNA damage accumulation [45]. In this context, the reduction in mitochondrial copy number in old *Nfu* skeletal muscle tissue (Figure 6e) was paralleled by a decreased expression of mitochondrial polymerase γ (Polγ), mitochondrial transcription factor A (Tfam), and mitochondrial single-stranded DNA-binding protein (Ssbp) (Figure 6f). 

Altogether, these results indicate that aged *Nfu* skeletal muscle is characterized by an increased inflammatory response associated with senescence and mitochondrial dysfunction.

## 4. Discussion

The epigenetic changes occurring during aging of *Nfu* skeletal muscle are poorly characterized. In the present study, we investigated changes of histone modifications and gene expression occurring upon age in the skeletal muscle of the turquoise killifish *Nfu.* Intriguingly, in aged *Nfu* skeletal muscle tissue, we observed a progressive increase of histone marks associated with heterochromatin (H3K27me3, H3K9me3, and H4K20me3), paralleled by a decrease in euchromatin histone marks (H3K9ac and H4K16ac). The ChIP-seq analysis for H3K27me3 and H3K9ac pointed out aging-dependent gene promoters epigenetically regulated. Down-regulated genes showed an enrichment of H3K27me3 associated with a relative decrease of H3K9ac, whereas up-regulated genes indicated a reduction of H3K27me3 in favor of increased levels of H3K9ac, suggesting for an opposite perhaps coordinated role of these two histone modifications during aging. Similar results were obtained in cells isolated from young and old *Nfu* skeletal muscle, suggesting this model as a potentially relevant in vitro system for the screening of senolytic and senomorphic compounds in an anti-aging perspective. In mammalian models of aging, conflicting observations have often been reported, including decrease in global histone and DNA methylation occurring with age paralleled by localized accumulation at specific CpG islands [21,46]. Indeed, the differences between *Nfu* and the more characterized mammalian models of aging may reflect the presence of general differences in the mechanisms determining epigenetic changes in mammalian and fish aging. Alternatively, this chromatin landscape could be a consequence of the very short lifespan of *Nfu*. More investigations would be necessary to elucidate this interesting aspect.

In the skeletal muscle of aged *Nfu,* gene expression analysis showed a reduction of genes involved in cell cycle and proliferation, including cyclins and PCNA. Similarly, the low expression of genes associated with DNA damage, compared to younger animals, suggests an accumulation of DNA damage with aging. Although DNA damage repair is essential for replication, proliferation, and tissue regeneration after injury, it seems impaired in old *Nfu* skeletal muscle as indicated by the accumulation of the modified γH2AX. Usually, in its natural habitat, killifish has only a short time period to grow and reproduce before the next dry season starts [47]. Possibly, this short lifespan compels *Nfu* to spend only little resources for DNA damage repair, a fundamental process for longer living species to counteract mutation and DNA damage accumulation [48]. Intriguingly, in other model organisms, genes involved in DNA damage repair have been shown to play essential roles in longevity, suggesting that genome maintenance in short-lived animals did not evolve as in long-lived species [48,49]. Insufficient DNA repair mechanisms could also explain the absence of lifespan extension in *Nfu* grown in fish tanks where food availability and lack of dry seasons represent optimal living conditions. On the other hand, *Nfu* might increase the proportion of heterochromatin with aging as a protective mechanism to prevent DNA breaks and other damages characterized by repetitive sequence silencing and transposon element inhibition [50]. In this light, several studies have already shown the impact of heterochromatin on lifespan and genome integrity maintenance. In *Drosophila,* heterochromatin formation contributes to longevity [51] and reduction of H4K16ac, via deletion of the Sas2 acetylase, apparently supports an increased lifespan [52]. In *Caenorhabditis elegans* (*C. elegans*)*,* the accumulation of H3K27me3 associates with increased lifespan and longevity, whereas high levels of H3K4me3 accelerates the aging process [53,54].

Aging-associated DNA damage partially depends on high levels of oxidative stress, which increase during lifespan, cell senescence, and aging-associated diseases [55,56]. Oxidative stress products accumulate in old skeletal muscle impairing mitochondrion function, which is essential for muscle contraction and strength [57]. Interestingly, it has been shown that locomotor activity is impaired in aged *Nfu* [31]. Indeed, old *Nfu* showed less swimming activity and mobility, suggesting that sarcopenia might occur with age [31]. However, further studies elucidating the physiological aspects and the connection between epigenetic changes and age-related alteration of skeletal muscle in *Nfu* would certainly be of interest. Sarcopenia, for example, is also associated with increased oxidative stress and decreased muscle function [58], which might partially depend on a decreased number of functional mitochondria. In this light, we observed a decline in mitochondrial copy number in old skeletal muscle tissue of *Nfu*, in agreement with previous analyses of other *Nfu* tissues [59]. Remarkably, the parallel reduction of mitochondrial proteins in *Nfu* skeletal muscle tissue could contribute to increase oxidative stress and ultimately cause DNA damage accumulation, cell cycle arrest, and senescence accelerating aging. In agreement, in old *Nfu* muscle tissue, we observed an association of increased levels of H3K9me3 with progressive accumulation of senescence markers. In particular, the so-called senescence-associated heterochromatin foci (SAHFs) were found enriched in regions where H3K9me3 and HP1α localized and chromatin was more condensed [15]. Consistent with an increase of senescent markers, we found the up-regulation of some inflammatory signaling pathways in old *Nfu* skeletal muscle tissue, which, becoming chronic, could possibly contribute to the so-called “inflammaging”, a matter that requires further future studies [60]. Among other signs of aging, we found an up-regulation of genes involved in glycolysis, a possible consequence of the typical metabolic switch occurring during aging [46]. 

While we could not always observe a linear increase or decrease of certain genes involved in the aging process, we would like to point out here that this specific aspect has been previously reported about gene expression in *Nfu* tissues during aging [61]. It has been hypothesized, in fact, that the so-called “U-shape curve” might be an intrinsic feature of *Nfu*. Interestingly, an inversion of gene expression at critical points of development and aging has also been reported in human and primates [62], where a U-shaped expression pattern has also been observed. We cannot exclude, however, that the small sample size and the consequential higher heterogeneity might itself affect the pattern. Moreover, the aged fish analyzed within this study might be the result of a positive selection of individuals presenting peculiar features about their gene expression [63]. 

Overall, the present work shed light on the epigenetic landscape of *Nfu* skeletal muscle during aging, pointing out features that might contribute to explain the loss of muscle mass, the reduction of cell proliferation, the increase of inflammation, the impairment of DNA repair, and the reduction in mitochondrial function. Interestingly, a recent study showed similar aging-associated transcription changes in *Nfu*, *Zebrafish*, mice, and humans [64]. Altogether, these observations further corroborate the use of *Nfu* as a suitable animal model for investigating the physiology and pathophysiology of aging. Nevertheless, some limitations in the present study are represented by a relatively small number of samples derived from aged fish and a lack of certain annotated marker genes in the genome of *Nfu*.

However, comparison with datasets derived from RNA-seq of other *Nfu* tissues revealed similar gene expression changes during aging. Specifically, the RNA-seq of skin, liver, and brain in old *Nfu* [61] shares at least six down-regulated and two up-regulated KEGG pathways with our RNA-seq conducted in skeletal muscle (Appendix A). Interestingly, in old *Nfu* skeletal muscle tissue, RNA-seq performed from Reichwald’s group [32] on diapause stage of *Nfu* showed a transcriptome similar to that observed in the present work (Appendix A). Although further investigation is necessary to elucidate this interesting finding, we can speculate that during diapause, the killifish embryo undergoes severe stress conditions, including aridity and temperature changes, which can contribute to the establishment of a transcriptome characterized by metabolic and cellular process shut-down similar to that observed during aging. 

The analysis of the epigenetic landscape of *Nfu* skeletal muscle during aging, performed in this study, provides an initial characterization of the epigenetic changes occurring during *Nfu* aging, which represents an interesting starting point of investigation to study aging and age-related alteration of skeletal muscle in a novel experimental animal model system.

## Figures and Tables

**Figure 1 cells-08-01169-f001:**
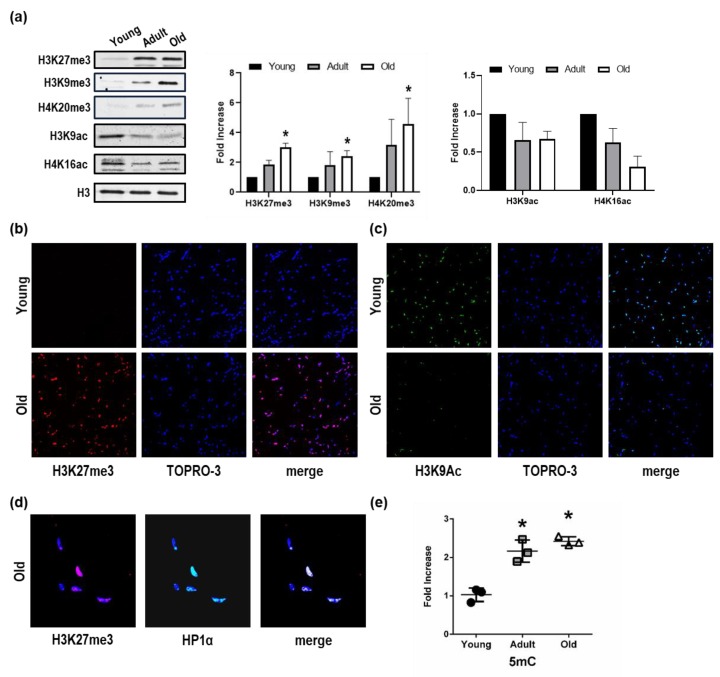
Repressive histone mark accumulation and activating histone mark decrease in *Nfu* skeletal muscle tissue with age. (**a**) Left panel: Representative Western blot analysis of H3K27me3, H3K9me3, H4K20me3, H3K9ac, and H4K16ac expression in young, adult, and old *Nfu* muscle tissue. Total histone 3 (H3) was used as loading control. Right panel: Related densitometry analysis of H3K27me3, H3K9me3, H4K20me3, H3K9ac, and H4K16ac expression in muscle tissue (*n* = 4). * *p* < 0.05 vs. young. (**b**) Representative confocal microscopy images of H3K27me3 (red) in young (upper panel) and old (lower panel) *Nfu* skeletal muscle tissue. Nuclei were counterstained with TOPRO-3 (blue). Magnification 40× (*n* = 5). (**c**) Representative confocal microscopy images of H3K9Ac (green) in young (upper panel) and old (lower panel) *Nfu* skeletal muscle tissue; nuclei were counterstained with TOPRO-3 (blue). Magnification 40× (*n* = 5). (**d**) Representative confocal microscopy images of H3K27me3 (red, left panel) and heterochromatin protein 1α (HP1α, green, middle panel) in old *Nfu* skeletal muscle tissue. Merged fluorescence images are shown in the right panel (merge). Nuclei were counterstained with TOPRO-3 (blue). Magnification 40× (*n* = 5). (**e**) Whole skeletal muscle global DNA quantification of 5-methylcitosine (5 mC) in young (black circles), adult (gray squares), and old (white triangles) *Nfu* skeletal muscle tissue expressed as fold-change versus young (*n* = 3). * *p* < 0.05 vs. young.

**Figure 2 cells-08-01169-f002:**
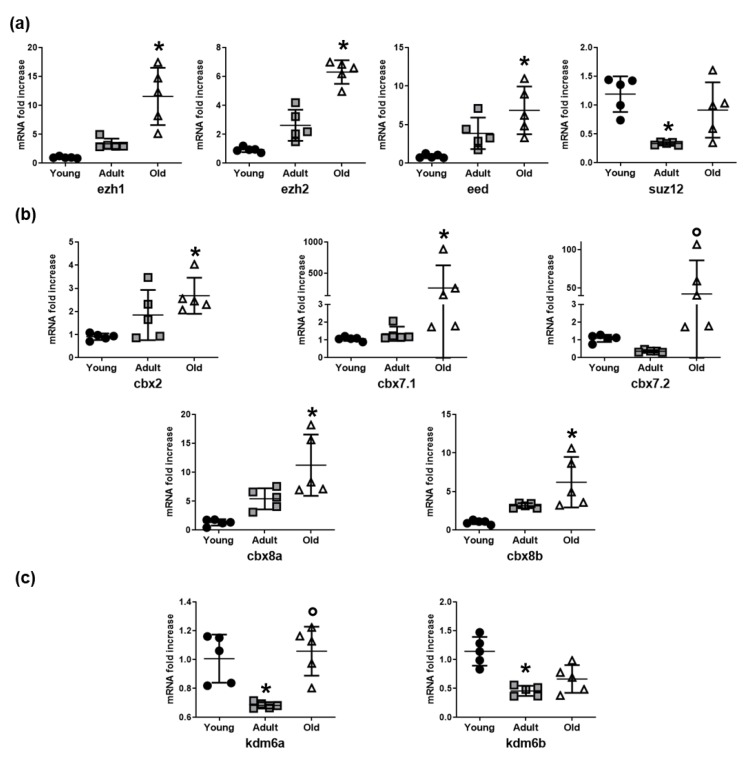
Histone methylation enzyme expression increases in *Nfu* muscle tissue with age. (**a**) qRT-PCR analysis of ezh1, ezh2, eed, and suz12, polycomb repressive complex 2 (PRC2) subunits in young (black circles), adult (gray squares), and old (white triangles) *Nfu* muscle tissue expressed as fold increase versus young samples (*n* = 5). * *p* < 0.05 vs. young. (**b**) qRT-PCR analysis of cbx2, cbx7.1, cbx7.2, cbx8a, cbx8b, a selection of polycomb repressive complex 1 (PCR1) mRNAs in young, adult, and old *Nfu* muscle tissue expressed as fold increase versus young samples. (n = 5). * *p* < 0.05 vs. young; *p* < 0.05 vs. adult. (**c**) qRT-PCR analysis of lysine demethylases kdm6a and kdm6b mRNAs in young, adult, and old *Nfu* muscle tissue expressed as fold increase versus young samples (*n* = 5). * *p* < 0.05 vs. young; *p* < 0.05 vs. adult.

**Figure 3 cells-08-01169-f003:**
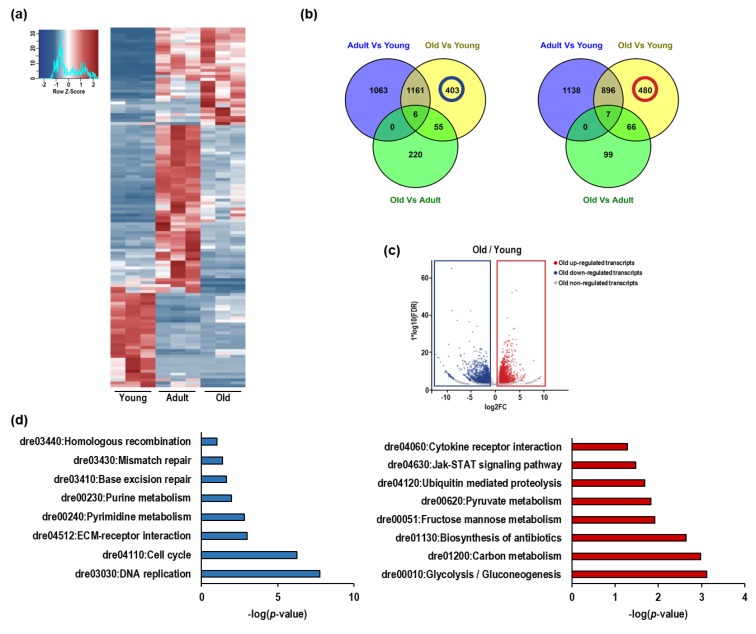
The expression of DNA repair and cell cycle genes decline in *Nfu* skeletal muscle with age, as expression of inflammation and glycolysis genes increase. (**a**) Heatmap showing the 50 most significant differentially regulated genes in young, adult, and old *Nfu* skeletal muscle identified by total RNA sequencing (*n* = 3 each group). Red and blue represent over- and under-expressed genes, respectively. (**b**) Left panel: Venn diagram depicting the distribution of unique or common down-regulated genes among young, adult, and old *Nfu* skeletal muscle tissue at different age. Right panel: Venn diagram depicting the distribution of unique or common up-regulated genes among young, adult, and old *Nfu* muscle. (**c**) Volcano plot of differentially regulated genes expressed in old *Nfu* skeletal muscle compared to young *Nfu* skeletal muscle at a multiple testing-corrected *p*-value ≤ 0.05. (**d**) Left panel: KEGG pathway over-representation analysis of unique down-regulated genes between young and old *Nfu* muscle tissue (405 genes), blue bar graph. Right panel: KEGG pathway analysis of unique up-regulated genes between young and old *Nfu* muscle tissue (473 genes), red bar graph.

**Figure 4 cells-08-01169-f004:**
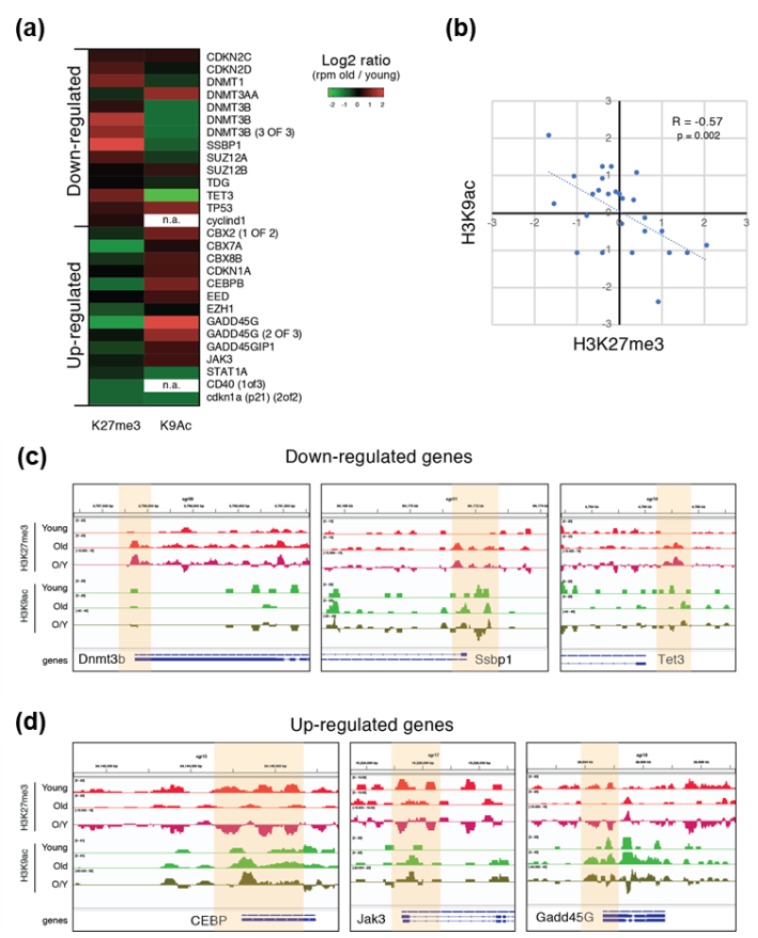
ChIP-seq reveals differential roles of H3K27me3 and H3K9ac in skeletal muscle tissue. (**a**) Heatmap showing the log2 fold change (old/young) of H3K27me3 and H3K9ac signal intensity (RPM) of some of the DEGs during aging (also see Appendix A). RPM = read per million. (**b**) Log2 fold change (old/young) of H3K27me3 (x axis) and H3K9ac (y axis) signal intensity (RPM). R and *p*-value is calculated using Pearson correlation. (**c**,**d**) Genomic view of the ChIP-seq-mapped reads for some DEGs which is down-regulated (**c**) or up-regulated (**d**) during aging.

**Figure 5 cells-08-01169-f005:**
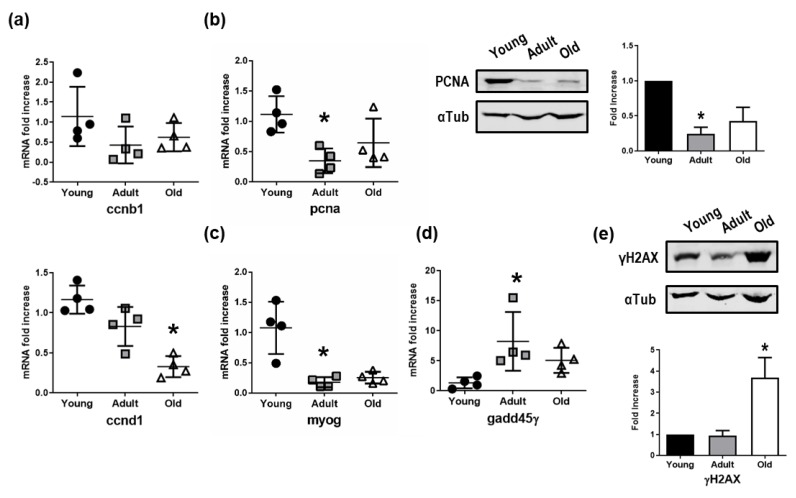
Validation of mRNA sequencing. (**a**) qRT-PCR analysis of cyclin B (ccnb1) and cyclin D (ccnd1) in young (black circles), adult (gray squares), and old (white triangles) *Nfu* muscle tissue expressed as fold increase versus young samples (*n* = 4). * *p* < 0.05 vs. young. (**b**) Left panel: qRT-PCR of the proliferating antigen (pcna), markers of cell cycle progression and proliferation. Middle panel: Representative Western blot analysis of PCNA protein levels in young, adult, and old *Nfu* muscle tissue. In each condition, α-tubulin (αTub) was used as loading control. Right panel: Related densitometry analysis of PCNA protein levels (*n* = 3); signal was normalized to young samples (black bars). Adult samples depicted with gray bar, old samples with white bars. * *p* < 0.05 vs. young. (**c**) RT-PCR analysis of myogenin-g (myog), the transcriptional activator of muscle differentiation in young (black circles), adult (gray squares), and old (white triangles) *Nfu* muscle tissue. (*n* = 4). * *p* < 0.05 vs. young. (**d**) RT-PCR analysis of growth and DNA damage inducible 45γ (gadd45γ) mRNAs in young (black circles), adult (gray squares), and old (white triangles) *Nfu* skeletal muscle tissue (*n* = 4). (**e**) Upper panel: Representative Western blot analysis of γ histone 2AX (γH2AX), a marker of DNA damage, in young, adult, and old *Nfu* muscle tissue. α-tubulin (αTub) was used as loading control. Lower panel: Related densitometry analysis of γH2AX in muscle tissue (*n* = 3).

**Figure 6 cells-08-01169-f006:**
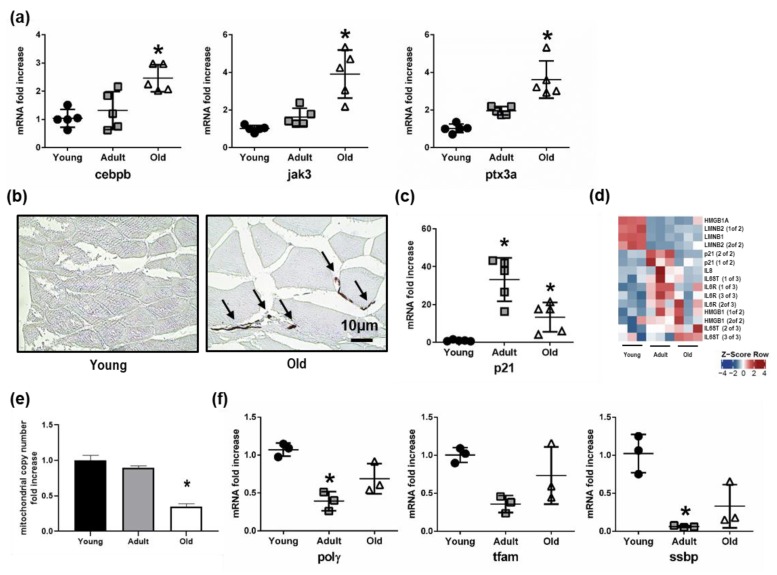
Senescence increases as mitochondrial number declines in the aging *Nfu* skeletal muscle. (**a**) Validation of sequencing results via RT-PCR analysis of mRNAs related to inflammatory signaling: CCAAT/enhancer binding protein beta (cebpb), janus kinase 3 (jak3), pentraxin 3a (ptx3a), in young (black circles), adult (gray squares), and old (white triangles) (*n* = 5). * *p* < 0.05 vs. young. (**b**) Representative light microscopic pictures from young (right) and old (left) *Nfu* muscle stained with Sudan Black B. Accumulation of staining in the old muscle is indicated with black arrows. Calibration bar = 10 µm. 40× magnification. (**c**) RT-PCR analysis of p21 mRNAs in young (black circles), adult (gray squares), and old (white triangles) *Nfu* skeletal muscle tissue (*n* = 5). * *p* < 0.05 vs. young. (**d**) Heatmap showing significantly differentially regulated senescent markers in young, adult, and old *Nfu* skeletal muscle identified by total RNA sequencing (*n* = 3 each group). Red and blue represent over- and under-expressed genes, respectively. (**e**) RT-PCR analysis of mitochondrial content number in young, adult, and old in *Nfu* muscle tissue expressed as fold increase versus young samples (*n* = 4). * *p* < 0.05 vs. young. (**f**) RT-PCR analysis of mitochondrial polymerase gamma (polγ), mitochondrial transcription factor (tfam) and mitochondrial single-stranded DNA binding protein (ssbp) in young, adult, and old *Nfu* muscle tissue expressed as fold increase versus young samples (*n* = 3 at each age). * *p* < 0.05 vs. young.

**Table 1 cells-08-01169-t001:** List of forward and reverse primers.

Gene	Forward 5’−3’	Reverse 5’−3’
cbx2	TCCCAACGGACAAAAGAAAC	TTGTTGGGTTTGGTGGATTT
cbx7.1	GAGCAAGTGTTTGCTGTGGA	CTTTGGCACCTTTCTTCCTG
cbx7.2	GGAGACAGGCTGGATTTTGA	GCCATGGTAACCGACTGATT
cbx8a	CAGTCAATCGGGGTGAAAGT	TTAGACTCCTCCGGGAACCT
cbx8b	AGGTGGCGAGTATCTGCTGT	CGGTTCCCAAGTGCTGTATT
ccnb	GGTGGGAGACTTTGCCTACA	AGAGGGTCAGCTCCATCAGA
ccnd	CTGTGAGCTTTGCTGCTTTG	ACGCTCAGCAAACACATACG
eed	AGTCCTGTGAAAACGCCATC	AACGTAAAGCTTCCCCACCT
ezh1	CAAGAGGATTCCCAGCGATA	GGGTTGGAGGAAACAGTCGTA
ezh2	ATTCTGTCAACCCCAACTGC	ATGCCCACGTACTTCAGAGC
gadd45γ	ATTGCGCTTCAGATCCACTT	CGCAGAACAGACTCAGCTTG
ira	TGCCTCTTCAAACCCTGAGT	AGGATGGCGATCTTATCACG
kdm6a	GTCAAACCCTACCCCCTCAT	TGTGGAGAGAGGAGCCAACT
kdm6b	CAAAGCCAGCTTTCTGGAAC	TCTGGATGTGAGGAGCACAG
myog	GTTCGACCAAGCTGGCTATC	CATGGTCACCGTCTTCCTTT
p21	CCCTGCGTAAAGATCTGGAG	ACCACCACCCTTCCTCTTT
pcna	ACCCTCAGAGCAGAGGACAA	CATGGGAAAGGATCTGGAAA
polγ	TCCCCGTTAATCAGAACTGG	TCTGCTGCTTTTTGGGAGTT
ssbp1	CTGGAGAGACGGAAACAAGC	CTGACGTTGTCGCTCAGAAA
suz12	AAAGGAGCAAAGGTGGAGGT	GACGGTTGTGACCACTGATG
tfam	TACGTGTCCGAGCACTTTCA	CATGTGGTCTTCCCAGGACT

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
