# Peer review of "Aging Triggers H3K27 Trimethylation Hoarding in the Chromatin of Nothobranchius furzeri Skeletal Muscle"

_cells, 2019, doi:10.3390/cells8101169_

Round 1

Reviewer 1 Report

The manuscript describes epigenetic and other molecular changes occurring in the skeletal muscle of Nothobranchius furzeri (Nfu) during aging. This short-lived fish is a relatively new model for aging studies.

The authors show that aged muscles of this fish (and cells isolated from aged fish) display a less accessible and more condensed chromatin characterized by a progressive increase of histone marks associated with heterochromatin and a decrease in histone marks associated with euchromatin. They also show reduced expression of genes involved in cell cycle, proliferation, DNA repair and dyfunctional mitochondria as well as an increase of DNA damage markers, inflammatory genes and cell senescence markers associated with aging. These observations suggest that Nfu is a suitable animal model for investigating the physiology and pathophysiology of aging.

The authors did not investigate a causal relationship or specific mechanisms but epigenetic changes occurring during aging of Nfu skeletal muscle have been scarcely investigated. Hence, this paper is an interesting starting point to study aging in a novel experimental animal model and may provide useful material for future studies of comparative genomic among species with different longevity.

I have only minor comments/suggestions for the authors.

Overall

The authors have focused a lot on sarcopenia. However, they did not perform functional assessment of sarcopenia in the fish that they have studied. They correctly state to investigate molecular/epigenetic changes occurring in aged muscles which may have implication for sarcopenia. Indeed, the observed changes occur normally during aging of the fishes and may not necessarily be implicated in sarcopenia. I would suggest to the authors to rephrase some sentences related to sarcopenia and focus better the discussion on age-related alteration of skeletal muscle.

Introduction

Prevalence of sarcopenia is around 10% in elderly above 60 years. The sentence stating that prevalence of sarcopenia is 42% in elderly with weaken muscle strength and impaired physical activity and movement appears rather “circular” and may confound the reader. I would suggest to report the correct prevalence of sarcopenia in elderly following the data from this paper (Shafiee et al. 2017, PMID: 28523252) or similar in literature. The causes of sarcopenia are still being investigated. Stating that “sarcopenia develops mostly as consequence of chronic inflammation and mitochondrial dysfunction” is too “strong”. I would suggest to say that “chronic inflammation and mitochondrial dysfunction have been shown to contribute to the development of sarcopenia”.

Discussion

The authors can provide more comparisons with epigenetic changes occurring in different species highlighting major differences and similarity of Nfu with other species. How do the authors interpret some U-shaped patterns? E.g. the p21 mRNA signal appears to be higher already in adult fish, data of poly, tfam and ssbp seems to display a U-shaped curve with aging. Is this a real pattern that may arise from inter-individual heterogeneity which leads to a positive selection of the population with higher physiological capacities (Piantanelli et al. 2001; PMID: 11470133)? Or is this simply the consequence of the low sample size?

At the end of the Discussion, review the limitations of your study. In my opinion the sample size may be a limit for some of the reported investigations where the experimental variability is high.

Author Response

The authors would like to thank the Reviewer for his/her useful comments to our work.

Overall

Q1. The authors have focused a lot on sarcopenia. However, they did not perform functional assessment of sarcopenia in the fish that they have studied. They correctly state to investigate molecular/epigenetic changes occurring in aged muscles which may have implication for sarcopenia. Indeed, the observed changes occur normally during aging of the fishes and may not necessarily be implicated in sarcopenia. I would suggest to the authors to rephrase some sentences related to sarcopenia and focus better the discussion on age-related alteration of skeletal muscle.

A1. We thank the Reviewer for his/her suggestion. Objective of our work is characterizing the molecular/epigenetic features of aging skeletal muscle. According to reviewer’s suggestion, in the revised version of the manuscript emphasis on sarcopenia has been reduced.

Introduction

Q2. Prevalence of sarcopenia is around 10% in elderly above 60 years. The sentence stating that prevalence of sarcopenia is 42% in elderly with weaken muscle strength and impaired physical activity and movement appears rather “circular” and may confound the reader. I would suggest to report the correct prevalence of sarcopenia in elderly following the data from this paper (Shafiee et al. 2017, PMID: 28523252) or similar in literature.

A2. We appreciate Reviewer’s comment about sarcopenia. In the revised manuscript, the Introduction has been modified accordingly (page 2) and the reference list updated accordingly (new ref 7).

Q3. The causes of sarcopenia are still being investigated. Stating that “sarcopenia develops mostly as consequence of chronic inflammation and mitochondrial dysfunction” is too “strong”. I would suggest to say that “chronic inflammation and mitochondrial dysfunction have been shown to contribute to the development of sarcopenia”.

A3. Our statement has been changed accordingly (page 2 lines 69-70).

Discussion

Q4. The authors can provide more comparisons with epigenetic changes occurring in different species highlighting major differences and similarity of Nfu with other species. How do the authors interpret some U-shaped patterns? E.g. the p21 mRNA signal appears to be higher already in adult fish, data of poly, tfam and ssbp seems to display a U-shaped curve with aging. Is this a real pattern that may arise from inter-individual heterogeneity which leads to a positive selection of the population with higher physiological capacities (Piantanelli et al. 2001; PMID: 11470133)? Or is this simply the consequence of the low sample size?

A4. We would like to thank the Reviewer raising the issue about U-shaped curves in the expression pattern of selected genes, including p21. We would like to point-out here that this specific aspect has been previously reported about gene expression in Nfu tissues during aging (Baumgart et al 2014). It has been hypothesized, in fact, that the U-shape curve might be an intrinsic feature of Nfu. Interestingly, an inversion of gene expression at critical points of development and aging has also been reported in human and primates (Cellerino and Ori 2017), where a U-shaped expression pattern has been also observed. We cannot exclude, however, that the small sample size and the consequential higher heterogeneity might affect the gene expression pattern. We agree with the Reviewer that aged fish we could analyze might be result of a positive selection of individuals with peculiar features about their genes expression. A specific comment about this important point has been included in the revised discussion section exploiting the suggested reference (page 16 lines 562-570; new ref 63).

Q5. At the end of the Discussion, review the limitations of your study. In my opinion the sample size may be a limit for some of the reported investigations where the experimental variability is high.

A5. We thank the Reviewer for this suggestion. A paragraph about the limitation of our study has been added in the revised discussion section (page 16 lines 571-579).

Reviewer 2 Report

Cencioni et al. studied aging in Killifish and showed modulations in histone methylation with age. Authors have used high throughput technique such as RNA seq and also CHIP seq to confirm the impairment in the cell cycle, differentiation, and DNA repair mechanism the may cause aging in muscles.   The paper has nicely portrayed the data that authors are conveying except the last section about senescence, which need some more data. The manuscript can be more useful if incorporated some more data.

My comments as follows,

Authors have mentioned that with age, a gradual accumulation of 5-methylcytosine (5mC) occurs in Nfu skeletal muscle tissue, but in figure 1e, there is no difference in adult and old at all. If aging does not cause the up-regulation, then it may be a significant contributor to aging. Line294, Authors talk about mRNA levels of kdm6a and kdm6b were reduced in adult. Authors are not describing the role of these proteins and why the modulation is essential? Also, for kdm6a, it appears that levels are similar in young and old. Authors have not addressed this as well. Figure-5, the names of the panels for mRNA fold increase should be changed to the scientifically accepted format. In the last section of the paper, authors are linking aging with senescence. Role of senescence is very well documented during aging, but authors do not have much evidence to support their claim. Classical markers of cellular senescence such as Lamin B1, p16, HMGB1, IL-6, IL-8, senescence-associated beta gal, etc. have not tested. Fig 6 c, p21 seems to be up-regulating in adult but going down in old. This is counterintuitive, as work in the field has shown precisely the opposite. Even though the link between inflammation, senescence, and aging is well known, authors have not demonstrated clear cut evidence that it is the phenomenon they are studying.

Author Response

The authors would like to thank the Reviewer for his/her useful comments to our work.

My comments as follows,

Q1. Authors have mentioned that with age, a gradual accumulation of 5-methylcytosine (5mC) occurs in Nfu skeletal muscle tissue, but in figure 1e, there is no difference in adult and old at all. If aging does not cause the up-regulation, then it may be a significant contributor to aging.

A1. We thank the Reviewer raising this important point. 5mC accumulates during Nfu lifespan. We do believe that accumulation in the adult stage might promote aging phenotype, as suggested by reviewer. Interestingly, we found that with age a steep increase of 5-methyl cytosine (5mC) occurs in Nfu skeletal muscle tissue at adult age compared to young muscle tissue. The level of 5mC remains high at old age, maintaining statistical significance against young samples and slightly increasing compared to adult skeletal muscle without reaching statistical significance (Figure 1e). These evidences suggest that the increase of 5mC at adult stage might promote aging phenotype and contribute to the establishment of heterochromatin. A specific comment has been added to the revised results section (Page 6 lines 259-267).

Q2. Line294, Authors talk about mRNA levels of kdm6a and kdm6b were reduced in adult. Authors are not describing the role of these proteins and why the modulation is essential? Also, for kdm6a, it appears that levels are similar in young and old. Authors have not addressed this as well.

A2. In our work, histone demethylase 6 was studied because of its specific role in H3K27me3 demethylation. According to our results, H3K27me3 seems extremely sensitive to aging. Hence, we studied whether there was any change in epigenetic enzymes involved in writing (Polycomb complex) and erasing (KDM) methylation groups from H3K27. This specific aspect has been addressed in the revised result section (Page 8 line 294-301).

Q3. Figure-5, the names of the panels for mRNA fold increase should be changed to the scientifically accepted format.

A3. As suggested, Cyclin B1 and Cyclin D1 names have been changed into CCNB1 and CCND1, respectively.

Q4. In the last section of the paper, authors are linking aging with senescence. Role of senescence is very well documented during aging, but authors do not have much evidence to support their claim. Classical markers of cellular senescence such as Lamin B1, p16, HMGB1, IL-6, IL-8, senescence-associated beta gal, etc. have not tested.

A4. The present study is aimed at investigating epigenetic marks of skeletal muscle aging in Nfu. Here, we investigated whether the expression of some genes more generally associated with senescence were also affected during the progression of Nfu life. We thank the Reviewer suggesting additional markers of senescence to consider. According our RNA sequencing data set, we found that Lamin B1 was down-modulated about -2.29 fold in old compared to young fish. This observation is in agreement with the reported reduction of Lamin B1 typically occurring during senescence (Freund A. MBOC 2012; Wang SA Sci Rep 2017). Notably, different from mammalians, in Nfu, p16 cannot be used as a senescence marker. Indeed, contrasting data are present in literature about p16 expression in teleosts. Previous manuscript, in fact, describe that no changes occur in p16 during aging (Graf M Experimental Geront 2013). For this reason, we focused on p21 expression. About other markers, HMGB1 is only slightly upregulated during aging of about 0.85 fold. Although IL-6 was not detected by our sequencing, its signalling transporter (IL6ST) and receptor (IL6R) were both significantly up-regulated during aging. Finally, IL8 was upregulated about 1.86 fold in old compared to young fish. These new data have been now reported in the revised version of Figure 6d as a new heatmap of senescence associated genes (Page 13).

We agree with the reviewer that Beta-gal staining well associates with senescence. However, this staining cannot be performed in formalin fixed para-embedded tissues. For this reason we took advantage of Sudan Black staining. Indeed, the detection of senescence-associated-beta-galactosidase activity (SA-β-gal), seems inapplicable to archival material. However, Georgakopoulou E.A. et al nicely validated the histochemical Sudan-Black-B (SBB) that specifically stains lipofuscin, an aggregate of oxidized proteins, lipids and metals, known to accumulate in aged tissues, and detectable independently of sample preparation (Georgakopoulou E.A. Aging 2013). In Figure 6b areas of Sudan Black positive cells can be appreciated into the skeletal muscle of old fish, whereas young samples, as expected, did not show any signal. A specific statement regarding the choice of Sudan Black staining has now been added in the revised result section (page 12-13 line 442-469).

Q5. Fig 6 c, p21 seems to be up-regulating in adult but going down in old. This is counterintuitive, as work in the field has shown precisely the opposite.

A5. Actually, p21 expression increases with statistical significance in adult and old fish compared to young samples. Instead, the apparent difference between adult and old is not statistically significant, p21 expression does not significantly change in old fish, rather its expression remains elevated since adulthood. This aspect is similar to the of 5mC, suggesting that the increase of p21 in adult fish might promote aging.

Q6. Even though the link between inflammation, senescence, and aging is well known, authors have not demonstrated clear cut evidence that it is the phenomenon they are studying.

A6. We agree with the reviewer. Indeed, “inflammaging” is not the topic of the present manuscript. Here, we are interested to the epigenetic characterization of Nfu age-related alteration of skeletal muscle. A specific statement clarifying this aspect has been added in the revised version of result section (page 12 line 439-442).

­

Round 2

Reviewer 2 Report

This manuscript is undoubtedly polished and nicer than previously submitted. Authors have done a lot of work, and I want to congratulate them for doing that. It was nice that the authors did some good work with the suggested senescence markers and quite impressed with that. I am still not convinced much with the reasoning provided by authors regarding kdm6a, p21, and 5mC. I do not see the link between those markers and aging. Levels of kdm6a and p21, in fact, going close to young individuals and certainly not aged. Authors are working out new model system and so it is always tricky in such situations.